# Application of Indocyanine Green Fluorescence Imaging in Assisting Biopsy of Musculoskeletal Tumors

**DOI:** 10.3390/cancers15082402

**Published:** 2023-04-21

**Authors:** Siyuan He, Ang Zhong, Jun Lei, Zhouming Deng, Xiaobin Zhu, Renxiong Wei, Huayi Huang, Zhenyi Chen, Lin Cai, Yuanlong Xie

**Affiliations:** Department of Spine Surgery and Musculoskeletal Tumor, Zhongnan Hospital of Wuhan University, Wuhan 430071, China

**Keywords:** bone and soft-tumor, optical biopsy, near-infrared fluorescence, indocyanine green

## Abstract

**Simple Summary:**

Biopsy and pathological diagnosis are the gold standard for the diagnosis of bone and soft tissue tumors. However, in the existing biopsy process, due to the heterogeneity of tumors, we find that there is still a high failure rate. As clinicians in the field of cancer, we hope to introduce the optical biopsy of bone and soft tissue tumors through the research into near-infrared fluorescence imaging in cancer. We utilize the accumulation effect of indocyanine green in tumors and use optical biopsy to further improve the success and reduce the failure rate of skeletal muscle tumor biopsy.

**Abstract:**

(1) Background: Biopsies are the gold standard for the diagnosis of musculoskeletal tumors. In this study, we aimed to explore whether indocyanine green near-infrared fluorescence imaging can assist in the biopsy of bone and soft tissue tumors and improve the success rate of biopsy. (2) Method: We recruited patients with clinically considered bone and soft tissue tumors and planned biopsies. In the test group, indocyanine green (0.3 mg/kg) was injected. After identifying the lesion, a near-infrared fluorescence camera system was used to verify the ex vivo specimens of the biopsy in real time. If the biopsy specimens were not developed, we assumed that we failed to acquire lesions, so the needle track and needle position were adjusted for the supplementary biopsy, and then real-time imaging was performed again. Finally, we conducted a pathological examination. In the control group, normal biopsy was performed. (3) Results: The total diagnosis rate of musculoskeletal tumors in the test group was 94.92% (56/59) and that in the control group was 82.36% (42/51). In the test group, 14 cases were not developed, as seen from real-time fluorescence in the core biopsy, and then underwent the supplementary biopsy after changing the puncture direction and the location of the needle channel immediately, of which 7 cases showed new fluorescence. (4) Conclusions: Using the near-infrared fluorescence real-time development technique to assist the biopsy of musculoskeletal tumors may improve the accuracy of core biopsy and help to avoid missed diagnoses, especially for some selected tumors.

## 1. Introduction

The World Health Organization’s recent classification of musculoskeletal tumors defines 117 different soft tissue tumors and 58 bone tumors [1,2]. Although the morbidity of musculoskeletal tumors is relatively low, which contrasts with other kinds of tumor such as liver cancer or lung cancer, they can lead to limb dysfunction and even death. Due to the increased number of subtypes of musculoskeletal tumors, sarcomas have a high degree of malignancy and poor prognosis, which bring great challenges to clinical diagnosis and treatment [3,4].

The key factors of cancer treatment are early diagnosis and early treatment [5]. Biopsy can help doctors to determine the pathological diagnosis of the tumor and then develop characteristic treatment plans. However, compared to other kinds of tumors, it is more difficult to make a precise pathological diagnosis of musculoskeletal tumors due to their diversity and the lack of specific biomarkers. At present, the pathological diagnosis of musculoskeletal tumors depends on the morphological characteristics of the tissues combined with clinical manifestations and imaging features [6]; therefore, qualified biopsy tissue specimens are a prerequisite for accurate pathological diagnosis. Bone tumors have deep lesions and are more difficult to puncture than prostate and breast cancers. At the same time, more musculoskeletal tumors have heterogeneous characteristics, which further increases the difficulty of obtaining accurate puncture specimens [7]. In the current biopsy process, there is no real-time method to determine whether the tumor biopsy lesions are successfully obtained to improve the accuracy rate of biopsy and then provide early detection, correct diagnosis, and timely treatment to patients.

Near-infrared (NIR) fluorescence imaging is a novel optical imaging technology that is more sensitive than direct visual inspection and palpation but can provide real-time guidance for tumor characteristic dissection [8]. Indocyanine green (ICG) is currently the only fluorophore approved by the US Food and Drug Administration (FDA) for intraoperative NIR imaging. After intravenous administration, ICG binds to serum proteins and appears as macromolecules in the circulation. ICG can passively accumulate in tumor tissue through enhanced permeability and retention (EPR) [9] and has been used to visualize tumors or sentinel lymph nodes in breast [10], gastric [11], lung [12], liver [13], and several other types of surgery [14,15]. Recently, some studies have reported the surgical resection of bone and soft tissue sarcomas with ICG-guided near-infrared imaging, which may prove that this technology can be safely used in the treatment of bone and soft tissue tumors [16,17]. However, near-infrared (NIR) fluorescence imaging for the detection of musculoskeletal tumor puncture specimens has not been reported. Few doctors stated that they could accurately obtain tumor tissue specimens in biopsy, and NIR imaging may provide us with a novel technique to improve the accuracy of musculoskeletal tumor biopsy.

In this study, we will try to develop a new method to test the efficacy of indocyanine green fluorescence imaging-assisted needle biopsy of musculoskeletal tumors. An appropriate amount of indocyanine green was administered intravenously 1 h before biopsy, and fluorescence imaging-assisted biopsy was performed during biopsy to determine whether the biopsy was effective. After searching a large amount of data, we did not find any relevant literature reporting on this method. We attempted to prove the effective auxiliary significance of fluorescence imaging in bone and muscle tumor biopsy through this clinical study in order to reduce the missed diagnosis of tumors through biopsy.

## 2. Materials and Methods

All subjects provided their informed consent for inclusion before they participated in this study. The study was conducted in accordance with the Declaration of Helsinki, and the protocol was approved by the Ethics Committee of the Zhongnan Hospital of Wuhan University (2022080) and is registered on chictr.org.cn (ChiCTR2200060540).

The main purpose of this study was to explore the feasibility and guiding significance of fluorescence navigation in core biopsy of bone and soft tissue tumors. All patients signed informed consent forms, all patients who underwent core biopsy had improved preoperative imaging examinations, including MRI, CT, and X-ray, and the imaging results were finally confirmed. Core biopsy was performed by an experienced orthopedic surgeon who had been trained in bone and soft tissue tumor biopsy for more than five years, and all pathological results were confirmed by more than two senior pathologists.

### 2.1. Patients and Groups

In this study, approximately 100 patients prepared for core biopsy were selected from 1 October 2021 to 1 October 2022. Inclusion criteria: 1. diagnosed as bone and soft tissue tumors using the imaging data (all the imaging was evaluated by the same two imaging physicians and one surgeon); 2. within the age range of 10–85 years old. Exclusion criteria: 1. imaging considered as non-tumor lesions; 2. diagnosed with infection or inflammation; 3. thyroid autonomic nodule, allergic to iodine or shellfish; 4. severe chronic kidney, liver, or lung diseases; 5. having the contraindications of cardiovascular and cerebrovascular system puncture. All patients were required to sign an informed consent form on a voluntary basis. The patients in the control group chose the normal core biopsy operation to obtain tissue. All the examination data, including MRI, CT, X-ray, and blood-tests, were obtained before the operation in both the test group and the control group. 

### 2.2. Core Biopsy Guided by Imaging Equipment

All patients were informed of the biopsy process and its potential complications before the proposed biopsy. In our treatment center, biopsies are usually performed under local anesthesia (1.0% lidocaine without adrenaline). In addition, a senior anesthesiologist examined the patient and participated in the entire operation to monitor the patient’s vital signs. All of these operations were performed by the same team with at least 5 years of experience in interventional bone oncology. All biopsies were performed in accordance with the biopsy procedure proposed by Francesco Traina et al. in the American Academy of Orthopedic Surgeons [6].

### 2.3. Research Design

In this study, a nonrandomized controlled trial was used to set up a control group and a test group, and all patients signed informed consent forms. The technical roadmap of the bone muscle tumor biopsy test group assisted by near-infrared fluorescence is shown in Figure 1.

#### 2.3.1. Control Group

All imaging examinations were performed before the operation. After the examination, the core biopsy was carried out according to normal biopsy operation. The patient underwent core biopsy under the guidance of B-ultrasound or CT equipment and was given a hemostatic bandage after operation. A sufficient number of samples were collected during the operation; the samples were soaked in formalin immediately after the operation, and the pathologist diagnosed the specimen tissue.

#### 2.3.2. Test Group

All imaging examinations were performed before the operation. According to the patient’s body weight, indocyanine green was injected intravenously at a dose of 0.3 mg/kg. One hour later, core biopsy was performed under the guidance of CT or B-ultrasound, and sufficient tissue was obtained during the operation. At the same time, the open fluorescence imaging system was used for fluorescence imaging. If there was no fluorescence development in the tissue obtained from the biopsy, we assumed that we had not acquired the tumor lesion successfully; consequently, the position of the puncture channel and biopsy tools were adjusted in the focus, the supplementary core biopsy was performed, and then the fluorescence development system was used for real-time fluorescence development. The non-fluorescence development and fluorescence development tissues were stored, and all the specimens were fixed with formalin. Then, the non-fluorescence development samples and fluorescence development samples were submitted for pathological examination, and the pathologist made the pathological diagnosis. Then, the effectiveness of this method was assessed using the consistency between the results of fluorescence development and the pathological results.

### 2.4. Investigational Drug

For this study, ICG (25 mg vial) was purchased from Ruida Pharmaceutical Co., Ltd. (Dandong, China). ICG is a kind of NIR fluorophore with a peak excitation wavelength of 805 nm, a peak emission wavelength of 830 nm, and a molecular weight of 775 Da. Indocyanine green (ICG) is a fluorescent group approved by the Food and Drug Administration (FDA) for intraoperative NIR imaging. The current price of ICG is around USD 1/mg in China.

### 2.5. Experimental Equipment

#### 2.5.1. Photo Device

The FloNavi^®^ 3100 Open Surgery Imaging System (Guangdong OptoMedic Technologies Inc., Foshan, China) was used to perform real-time surgical macro fluorescence imaging at the operating table. The system is equipped with a handheld camera and a high-precision dual-frequency (white and near-infrared) camera system capable of emitting and detecting light in the NIR spectrum. For NIF imaging, four sub-windows on the monitor display images in different modes, including white light, original fluorescence, green fluorescence, and color fluorescence. The fluorescence imaging system was provided for free for this study.

#### 2.5.2. Biopsy Device

CT navigation equipment was provided by Siemens dual-source CT, equipment model: Force. The B-mode ultrasonic equipment was the LOGIQ produced by the GE Company of the United States, and the model is E11.The bone lesion biopsy tool was provided by Wuhan Yijiabao Biological Materials Co., Ltd. (Wuhan, China), and the model is ZT-CC-07. The soft tissue biopsy tool was provided by the AMERICAN Becton, Dickinson and Company (East Rutherford, NJ, USA), and the model is MC1816.

### 2.6. Effectiveness of Fluorescence-Assisted Biopsy

The diagnostic efficiency and accuracy for bone and soft tissue tumors in the experimental group and the control group were analyzed to evaluate the effectiveness of fluorescence-assisted biopsy. After the operation, the pathological results were collected and confirmed with a pathologist.

### 2.7. Data Analysis

Fisher’s exact test, the *t* test, and one-way ANOVA were used to compare between groups. IBM SPSS Statistics, version 26.0 (IBM Corporation, Armonk, NY, USA), was used for data collection and analysis. *p* < 0.05 was considered to be significant.

## 3. Results

In this study, we collected a total of 140 cases with suspected bone and soft tissue tumors by medical imaging, and all patients signed biopsy consent forms and informed consent forms for clinical trials. Finally, in the test group, we carried out near-infrared fluorescence-assisted musculoskeletal tumor biopsy in a total of 71 patients according to the inclusion and exclusion criteria. After fluorescence-assisted biopsy, 12 patients displayed inflammation in the final pathological examination and were excluded from this clinical trial, in accordance with the exclusion criteria. The final diagnosis of bone and soft tissue tumors included 59 cases. In the control group, there were 58 patients, of which 2 patients decided to withdraw. After routine biopsy, five patients were diagnosed with infection and inflammation and excluded from this clinical trial; these patients had also been cured after anti-infection treatment. Fifty-one cases were finally diagnosed as bone and soft tissue tumors. The flowchart of this trial is shown in Figure 2.

In the test group, there were 9 cases of primary benign tumors (15.25%), 25 cases of primary malignant tumors (42.38%), 16 cases of metastatic tumors (27.12%), 6 cases of tumor-like lesions (10.17%), and 3 cases of missed diagnoses (5.08%). In the control group, there were 10 cases of primary benign tumors (19.61%), 25 cases of primary malignant tumors (49.02%), 4 cases of metastatic tumors (7.84%), 3 cases of tumor-like lesions (5.89%), and 9 cases of missed diagnoses (17.64%). The characteristics of patients and biopsy sites in the test group and the control group are shown in Table 1. There was no significant difference in clinical and demographic characteristics between the two groups (Table 1, *p* > 0.05). The final pathological results of each biopsy guidance group after surgical resection of the tumor are shown in Table 2. In terms of postoperative pathological types, there was no significant difference between the test group and the control group (Table 2, *p* > 0.05).

To directly compare the results of musculoskeletal biopsy with the results of the gold standard examination, we evaluated the overall diagnostic efficiency and accuracy of the test group and the control group in the diagnosis of musculoskeletal tumors (Table 3). The total diagnosis rate of musculoskeletal tumors in the test group was 94.92% (56/59) and that in the control group was 82.36% (42/51). The positive rate of diagnosis in the test group (94.92%) was higher than that in the control group (82.36%) (*p* < 0.05). Among the 51 cases in the control group, 9 cases were dissatisfactory because there were no tumor cells (1 case of osteosarcoma, 1 case of fibrosarcoma, 1 case of Ewing’s sarcoma, 2 cases of metastatic tumor) and a large amount of necrotic tissue (2 cases of osteosarcoma and 2 cases of liposarcoma). The pathological results of the other 42 lesions were entirely consistent with the results of the operation and pathology. Of the 59 lesions in the test group, we considered that the puncture results were not satisfactory in 3 cases (3.59%) due to the absence of tumor cells (2 cases of metastatic tumor and 1 case of liposarcoma). The pathological results of the other 56 musculoskeletal tumors were consistent with those of the operation and pathology. At the same time, we sorted the fluorescence development results and characteristics of all patients who underwent indocyanine green near-infrared fluorescence-assisted biopsy, as shown in Table 4.

For 14 cases (23.73%) without fluorescence development of the first puncture, we assumed that we had not acquired the tumor lesions, and then the puncture channel and direction were adjusted immediately and real-time fluorescence development after the supplementary biopsy was performed in 7 cases (11.86%). Among these seven cases (11.86%) for whom supplementary puncture fluorescence was not developed, three cases (two cases of metastatic tumor and one case of liposarcoma) showed no tumor cells in the biopsy and were confirmed as missed diagnoses. Pathological examination confirmed a malignant tumor after open surgical resection of the tumor. Although the four other patients did not undergo real-time fluorescence development twice, the final pathological examination confirmed tumor-like lesions (two cases of bone fibrous dysplasia and one case of bone cyst) and primary benign tumors (one case of soft tissue myxoma). We sorted out the characteristics of 14 cases in the experimental group whose first puncture specimens were not developed, as shown in Table 5.

In the group of musculoskeletal tumor biopsies assisted by near-infrared fluorescence, we listed three typical cases. The results of the patients who underwent real-time fluorescence imaging after the first biopsy are shown in Figure 3. In these patients, the fluoroscopic tissue after the first biopsy was finally confirmed as a tumor by pathologists. Figure 4 and Figure 5 show the results of the supplementary biopsy tissue real-time fluorescence development in patients after the real-time fluorescence development of the first biopsy tissue was not developed and the direction and position of the biopsy channel were adjusted immediately. In these patients, the fluorescence of the undeveloped tissue was finally confirmed as a non-tumor component, such as adipose or inflammatory tissue, by pathologists after the first biopsy, and the tumor component was finally confirmed by pathologists after the supplementary biopsy.

## 4. Discussion

Biopsy has been found to be a safe, reliable, and accurate diagnostic tool in the diagnosis of musculoskeletal tumors. However, the positive rate of puncture biopsy of musculoskeletal tumors reported in the literature varies (70–97%) in different hospitals [7,18,19,20,21,22], and it is still necessary to improve the accuracy of biopsy. Near-infrared fluorescence imaging based on ICG has been used in tumor surgery by using clathrin-mediated endocytosis and the EPR effect of indocyanine green [23,24,25]. In 2021, studies by Brookes MJ [16] and Nicoli F [17] seem to have confirmed the feasibility of indocyanine green in bone and soft tissue tumors. Additionally, researchers have applied indocyanine green in biopsy, such as in lymph node biopsy of gastric cancer [26] and breast cancer [27]. In our study, wed use near-infrared fluorescence based on ICG to assist biopsy in real time, so as to explore the feasibility and application value of this new method of biopsy.

In the study of biopsy of bone and soft tissue tumors, although many researchers used different methods to improve the positive rate of biopsy, such as PET-CT-guided biopsy [28] and liquid biopsy [29], and the positive rate of biopsy has increased significantly, there is still a risk of biopsy failure. Even with the most advanced and accurate CT navigation or B-mode ultrasound equipment, a 100% success rate and diagnostic accuracy of core biopsy cannot be guaranteed. The reasons for biopsy failure may be related to the sample size of the puncture, the characteristics of the tumor lesions, infection, bleeding, necrotic tissue, and the patient’s personal reasons [7]. Recently, optical biopsy has become a research area of interest. Muriel Abbaci [30] et al. introduced confocal laser microscopy (CLE), which can provide real-time histological visualization of living tissue. In our study, we found that the positive rate of biopsy in the test group was significantly higher than that in the control group, and the positive rate (94.92%) seemed to maintain a similarly high level to that reported in the literature. Here, we show the results of three typical cases we studied (Figure 3, Figure 4 and Figure 5). We found that near-infrared fluorescence imaging has a certain auxiliary significance in the biopsy of bone and soft tissue tumors.

At the same time, in our study, we found that in the test group, there were 11 cases of homogeneous tumors and 48 cases of heterogeneous tumors in MRI imaging, and the positive rates of biopsies were 100% and 93.75%, respectively. In seven patients whose specimens were successfully obtained after real-time fluorescence development of supplementary biopsies, the MRI findings were also heterogeneous. Additionally, in one case of bone metastases with sclerosing changes on CT, obtain tumor cells failed to be obtained. In general, not all tumors require biopsy to be conducted [7,31], and the results of biopsies are often affected by many factors, such as tumor characteristics and tumor size [32]. For malignant tumors, the positive rate of biopsy was higher in tumors with a higher grade and more metastatic and lytic components, while it was lower in tumors with sclerosing and small or major necrotizing lesions [7]. In fact, near-infrared fluorescence imaging based on ICG may be more helpful in some tumors with characteristics including heterogeneous tumors in MRI and smaller size, etc. In these tumors, near-infrared fluorescence imaging may effectively distinguish necrotic tissue or normal tissue from tumor tissue, thus improving the accuracy of biopsy. 

More studies have shown that the diagnostic accuracy of percutaneous biopsy has not decreased compared to open biopsy [33,34], and the risk of complications from percutaneous biopsy is lower. The disadvantages of open biopsy are its high cost and high incidence of complications, including hematoma and tumor dissemination. Therefore, image-guided percutaneous core biopsy or fine-needle aspiration have become effective alternative methods [35]. However, open biopsy is extremely valuable in certain special cases, especially for the diagnosis of some difficult and complex diseases. Open biopsy is usually applied when the diagnosis after percutaneous biopsy is uncertain or unrelated to clinical manifestations and radiological results [36], or used in biopsies of specific sites [6]. In our study, there were three cases in the test group for which tumor cells were not found after core biopsy, resulting in missed diagnoses (these three cases did not develop fluorescence twice), and similar conditions also existed in the control group. Finally, after the core biopsy failed, we performed open biopsy on these cases and obtained tumor samples. Overall, NIR imaging has certain potential value in the biopsy process of musculoskeletal tumors. Due to the lack of evidence that one biopsy technique is superior to another, scientists are encouraged to explore biopsy techniques with higher diagnostic accuracy.

It is also reported that some researchers used pathological frozen sections to determine whether or not it is effective to obtain tumor samples during biopsy. For example, in the biopsies of uterine tumors [37] and bone tumors [38], frozen section pathological examination can help to determine the effectiveness of biopsies, thus improving the positive rate of the biopsies. However, intraoperative frozen pathological sections still have certain time limits and errors. In the study of Wallace MT et al., intraoperative frozen sections of bone tumors needed to wait for an hour [39], while in the Sezak M study, five patients had delayed biopsies because there were no results in the pathological frozen sections [40]. In the process of soft tissue tumor biopsy, Miwa S et al. have shown that frozen section pathology is not effective in the process of soft tissue tumor biopsy [41]. At the same time, in some developing countries and underdeveloped areas, due to the lack of pathological laboratory conditions, it may not be possible to support the implementation of pathological frozen sections in biopsies. Additionally, there may be some difficulties in using frozen sections due to the particularity of the bone structure. Therefore, a real-time and rapid method of biopsy diagnosis is needed. In our exploratory study, indocyanine green near-infrared fluorescence development may assist the rapid biopsy of bone and soft tissue tumors and in real-time, so as to improve the accuracy of biopsy and avoid missed diagnosis.

This new method may have a good clinical prospect in the diagnosis of bone and soft tissue tumors, but more sample data are still needed to support the research. In biopsy operations, a near-infrared fluorescence photography system can be used to determine whether or not the tumor tissue can be obtained effectively and rapidly in real time. The use of this new technique may improve the accuracy of tumor biopsy, but the indication of this method needs further exploration and development. At the same time, we are also trying to find more specific fluorescent dyes and targeted fluorescent molecular groups so that fluorescence navigation may play a more important role in tumor biopsy and surgical resection in the future.

## 5. Conclusions

Near-infrared (NIR) fluorescence imaging may assist the biopsy of bone and soft tissue tumors. This clinical study of indocyanine green-assisted biopsy of musculoskeletal tumors may help us develop a new method of biopsy, which may help surgeons reduce some of the difficulties of biopsy operations and improve the collection rate of tumor specimens, ultimately benefiting patients. Additionally, this new method of biopsy may be helpful in some selected tumors.

## Figures and Tables

**Figure 1 cancers-15-02402-f001:**
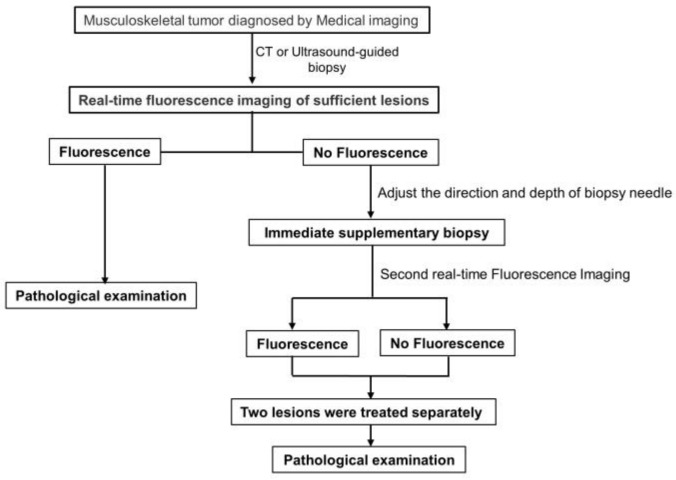
Research and design technology roadmap of the test group.

**Figure 2 cancers-15-02402-f002:**
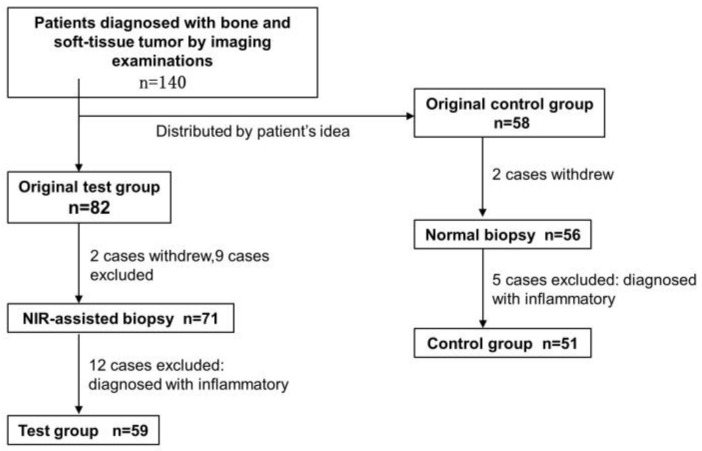
The flowchart of this trial.

**Figure 3 cancers-15-02402-f003:**
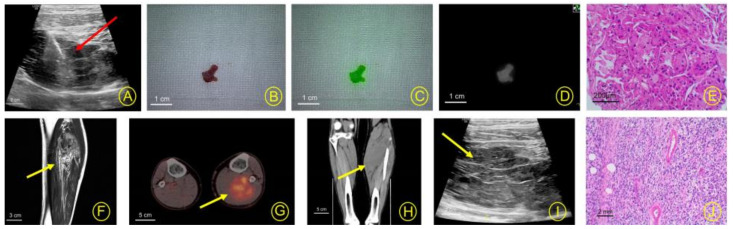
Thirty-two-year-old male suspected to have malignant tumors of the lower limb soft tissue. (**A**) Under the guidance of B-ultrasound, the biopsy needle was implanted into the lesion (red arrow). (**B**–**D**) The naked-eye view of the biopsy lesions, the green fluorescence, and original fluorescence of the real-time fluorescence development of the biopsy lesions. (**E**) The histopathologic biopsy results (hematoxylin and eosin, original magnification 100×) confirmed the bone lesion as a non-Hodgkin’s lymphoma. (**F**–**I**) Medical imaging before biopsy for the diagnosis of soft tissue tumors (yellow arrow). (**J**) The pathological results (non-Hodgkin’s lymphoma) after surgical resection of the humeral tumor were consistent with the pathological results of the biopsy (hematoxylin and eosin, original magnification 40×).

**Figure 4 cancers-15-02402-f004:**
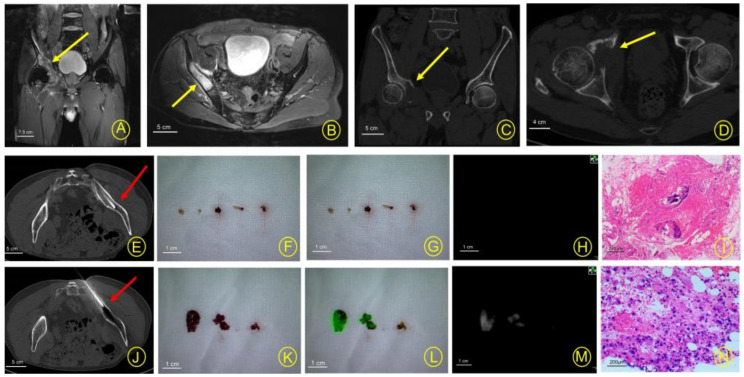
A 35-year-old male suspected to have malignant tumors of the right pelvis and surrounding soft tissue. (**A**–**D**) Medical imaging before biopsy for the diagnosis of pelvic malignant tumors (yellow arrow). (**E**) Under the guidance of CT, the biopsy needle was implanted into the lesion (red arrow). (**F**–**H**) Naked-eye view of biopsy lesions and real-time fluorescence imaging of lesion specimens (no fluorescence imaging). (**I**) Pathological results of the first biopsy (necrotic lesions, inflammatory tissues). (**J**) The position and direction of the puncture channel were immediately adjusted, the supplementary biopsy was performed (under the guidance of CT), and then real-time fluorescence imaging was performed again after the lesions were obtained (red arrow). (**K**–**M**) The naked-eye view of the biopsy lesions, the green fluorescence, and original fluorescence of the real-time fluorescence development of the supplementary biopsy lesions. (**N**) The histopathologic biopsy results (hematoxylin and eosin, original magnification 100×) confirmed the bone lesion as a fibrosarcoma.

**Figure 5 cancers-15-02402-f005:**
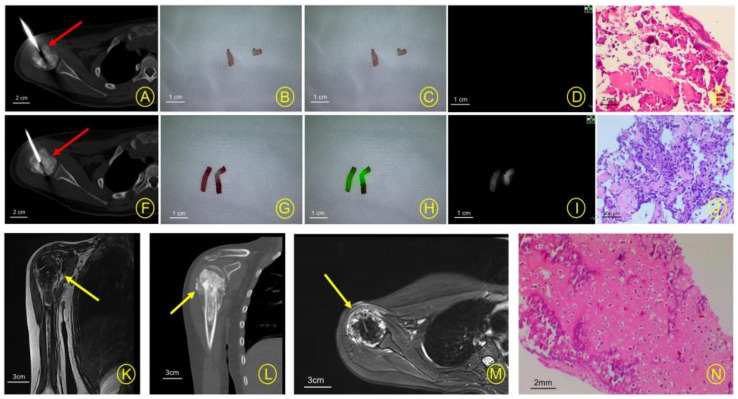
(**A**) Under the guidance of CT, the biopsy needle was implanted into the lesion (red arrow). (**B**–**D**) The naked-eye view of the biopsy lesions and real-time fluorescence imaging of the lesion specimens (no fluorescence imaging). (**E**) Pathological results of the first biopsy (necrotic tissue). (**F**) The direction and position of the needle track were adjusted immediately, and the focus was obtained again. After the end of the biopsy, the specimen was developed with real-time fluorescence. In the picture, the biopsy needle was implanted into the lesion (red arrow). (**G**–**I**) The naked-eye view of the biopsy lesions, the green fluorescence, and original fluorescence of the real-time fluorescence development of the supplementary biopsy lesions. (**J**) The pathological results of the supplementary biopsy (osteosarcoma). (**K**–**M**) Imaging data before biopsy (yellow arrow). (**N**) The pathological results (osteosarcoma) after surgical resection of the humeral tumor were consistent with the pathological results of the biopsy.

**Table 1 cancers-15-02402-t001:** Comparison of the baseline clinical characteristics of the patients in the test and control groups.

Variable	Test Group *n* = 59	Control Group *n* = 51	χ^2^ or t Value	*p* Value
Sex				
Male	37	28	0.690	0.406
Female	22	23		
Age (mean + SD)	53.61 ± 19.05	39.53 ± 17.44		
Localization of the lesion biopsy				
Spinal column				
Thoracic vertebrae	1	1	0.011	0.917
Lumbar vertebrae	2	1	0.016	0.898
Sacrum	2	2	0.131	0.717
Upper extremity				
Humerus	3	5	0.339	0.560
Ulna	1	1	0.011	0.917
Upper soft tissue	5	4	0.052	0.819
Lower extremity				
Femur	12	12	0.163	0.686
Tibia	2	3	0.028	0.867
Calcaneus	2	1	0.016	0.898
Lower soft tissue	13	9	0.329	0.566
Pelvis	8	8	0.100	0.752
Trunk	8	4	0.920	0.338
Postsurgical histopathology diagnosis				
Primary benign tumor	9	10	0.363	0.547
Primary malignancies	25	26	0.815	0.367
Metastases	19	12	1.017	0.313
Tumor-like lesions	6	3	0.220	0.639
Biopsy guidance method				
CT guidance	36	34	0.377	0.539
Ultrasound aguidance	23	17		

**Table 2 cancers-15-02402-t002:** Comparison of the pathological findings after open surgery between the test and control groups.

Postsurgical Histopathology Diagnosis	Test Group *n* = 59	Normal Groups *n* = 51	χ^2^ Value	*p* Value
Primary benign tumors				
Chondroma	2	3	0.028	0.867
Hemangioma	1	2	0.016	0.898
Lipoma	1	2	0.016	0.898
Schwannoma	3	1	0.131	0.717
Giant-cell tumor	1	1	0.011	0.917
Myxoma	1	1	0.011	0.917
Primary malignancies				
Liposarcoma	4	4	0.024	0.878
Lymphoma	4	2	0.056	0.812
Fibrosarcoma	3	3	0.056	0.812
Rhabdomyosarcoma	1	1	0.011	0.917
Leiomyosarcoma	1	3	0.435	0.510
Chondrosarcoma	1	1	0.011	0.917
Osteosarcoma	9	10	0.363	0.547
Ewing sarcoma	1	2	0.016	0.898
Granulosa cell tumor	1	0	0.373	0.542
Metastases				
Lung tumor	6	4	0.008	0.928
Breast tumor	5	3	0.024	0.878
Kidney tumor	2	1	0.016	0.898
Prostate tumor	2	1	0.016	0.898
Stomach tumor	1	0	0.373	0.542
Uterus tumor	2	3	0.028	0.867
Skin tumor	1	0	0.373	0.542
Tumor-like lesions				
Fibrous dysplasia of bone	3	0	1.094	0.296
Aneurysmal bone cyst	1	2	0.016	0.898
Simple cyst	2	1	0.016	0.898

**Table 3 cancers-15-02402-t003:** Comparison of the diagnostic performance between the test and control groups.

	Biopsy Group		χ^2^ Value	*p* Value
	Test Group	Control Group		
Overall diagnostic yield	94.92% (56/59)	82.36% (42/51)	4.442	0.035
Diagnostic accuracy	94.92% (56/59)	82.36% (42/51)	4.442	0.035

**Table 4 cancers-15-02402-t004:** Characteristics and effect of real-time fluorescence development in the test group.

Number	Age	Sex	Location	Max Dimension (mm)	Operation Time	ICG Dose	Time of Administration	First Fluorescent	Second Fluorescent	Bleeding Volume (mL)	Biopsy Dignosis	Open Surgey Dignosis
1	44	M	Femur	45	0.5 h	0.3 mg/kg	3 h pre-op	yes	no	20	Tumor-like lesions	Tumor-like lesions
2	32	M	Lower soft tissue	86	1 h	0.3 mg/kg	1 h pre-op	yes	no	5	Malignancies	Lymphoma
3	53	M	Trunk	156	0.5 h	0.3 mg/kg	1.5 h pre-op	no	Yes	10	Malignancies	Granulosa cell tumor
4	61	F	Trunk	55	1 h	0.3 mg/kg	1 h pre-op	yes	no	20	Malignancies	Osteosarcoma
5	69	F	Femur	27	0.8 h	0.3 mg/kg	1 h pre-op	yes	no	20	Malignancies	Metastases
6	21	M	Femur	49	1 h	0.3 mg/kg	1 h pre-op	yes	no	40	Malignancies	Osteosarcoma
7	50	F	Lower soft tissue	165	0.5 h	0.3 mg/kg	1 h pre-op	yes	no	8	Malignancies	Osteosarcoma
8	50	F	Upper soft tissue	91	0.5 h	0.3 mg/kg	1.5 h pre-op	no	no	5	Non-tumor	Metastases
9	68	M	Ulna	39	1 h	0.3 mg/kg	1 h pre-op	yes	no	21	Malignancies	Metastases
10	52	F	Lower soft tissue	125	0.5 h	0.3 mg/kg	1.5 h pre-op	no	no	10	Non-tumor	Liposarcoma
11	73	F	Spinal column	39	1 h	0.3 mg/kg	1.5 h pre-op	yes	no	6	Malignancies	Metastases
12	64	F	Spinal column	46	1 h	0.3 mg/kg	1.5 h pre-op	no	no	10	Non-tumor	Metastases
13	45	F	Spinal column	42	1 h	0.3 mg/kg	1 h pre-op	yes	no	5	Malignancies	Metastases
14	65	M	Femur	28	0.8 h	0.3 mg/kg	1.5 h pre-op	no	no	12	Tumor-like lesions	Tumor-like lesions
15	45	M	Tibia	54	1 h	0.3 mg/kg	1 h pre-op	yes	no	15	Malignancies	Osteosarcoma
16	64	M	Upper soft tissue	48	1 h	0.3 mg/kg	1 h pre-op	yes	no	15	Malignancies	Osteosarcoma
17	68	M	Calcaneus	31	0.5 h	0.3 mg/kg	1 h pre-op	yes	no	15	Malignancies	Osteosarcoma
18	80	M	Upper soft tissue	48	0.5 h	0.3 mg/kg	1.5 h pre-op	yes	no	5	Malignancies	Osteosarcoma
19	18	M	Femur	54	1 h	0.3 mg/kg	1.5 h pre-op	yes	no	15	Malignancies	Osteosarcoma
20	62	F	Pelvis	26	1 h	0.3 mg/kg	1.5 h pre-op	no	no	10	Tumor-like lesions	Tumor-like lesions
21	67	M	Pelvis	100	1 h	0.3 mg/kg	1 h pre-op	yes	no	20	Malignancies	Metastases
22	71	M	Pelvis	14	1 h	0.3 mg/kg	1.5 h pre-op	yes	no	16	Malignancies	Metastases
23	52	M	Humerus	26	0.8 h	0.3 mg/kg	1.5 h pre-op	yes	no	5	Benign tumor	Chondroma
24	66	F	Spinal column	36	1 h	0.3 mg/kg	1.5 h pre-op	yes	no	5	Benign tumor	Bone hemangioma
25	23	M	Calcaneus	49	0.5 h	0.3 mg/kg	1.5 h pre-op	yes	no	16	Tumor-like lesions	Tumor-like lesions
26	73	F	Trunk	84	0.5 h	0.3 mg/kg	2 h pre-op	yes	no	5	Malignancies	Lymphoma
27	60	M	Pelvis	32	1 h	0.3 mg/kg	2 h pre-op	yes	no	20	Malignancies	Metastases
28	51	M	Femur	86	1 h	0.3 mg/kg	1.5 h pre-op	yes	no	15	Malignancies	Metastases
29	58	F	Pelvis	36	1 h	0.3 mg/kg	1.5 h pre-op	no	Yes	10	Malignancies	Metastases
30	60	M	Femur	45	2 h	0.3 mg/kg	2 h pre-op	yes	no	15	Malignancies	Metastases
31	50	F	Tibia	36	1 h	0.3 mg/kg	1.5 h pre-op	no	no	10	Tumor-like lesions	Tumor-like lesions
32	35	M	Pelvis	186	1 h	0.3 mg/kg	2 h pre-op	no	Yes	10	Malignancies	Bone fibrosarcoma
33	60	M	Femur	48	1 h	0.3 mg/kg	1.5 h pre-op	yes	no	10	Malignancies	Metastases
34	75	F	Lower soft tissue	66	1 h	0.3 mg/kg	2 h pre-op	no	Yes	10	Malignancies	Rhabdomyosarcoma
35	59	F	Upper soft tissue	78	0.5 h	0.3 mg/kg	1.5 h pre-op	yes	no	5	Benign tumor	Lipoma
36	77	M	Femur	38	1 h	0.3 mg/kg	2 h pre-op	yes	no	15	Malignancies	Metastases
37	78	F	Lower soft tissue	128	0.5 h	0.3 mg/kg	1.5 h pre-op	yes	no	5	Malignancies	Lymphoma
38	73	F	Trunk	206	0.5 h	0.3 mg/kg	2 h pre-op	yes	no	3	Benign tumor	Schwannoma
39	17	M	Lower soft tissue	65	0.5 h	0.3 mg/kg	2 h pre-op	yes	no	5	Malignancies	Lymphoma
40	17	F	Humerus	46	0.5 h	0.3 mg/kg	2 h pre-op	yes	no	10	Benign tumor	Bone giant-cell tumor
41	60	M	Lower soft tissue	102	0.5 h	0.3 mg/kg	1.5 h pre-op	no	Yes	5	Malignancies	Leiomyosarcoma
42	61	M	Lower soft tissue	34	1 h	0.3 mg/kg	1.5 h pre-op	yes	no	5	Malignancies	Metastases
43	41	M	Femur	55	1 h	0.3 mg/kg	2 h pre-op	yes	no	10	Malignancies	Chondrosarcoma
44	81	F	Pelvis	61	1 h	0.3 mg/kg	1.5 h pre-op	yes	no	5	Tumor-like lesions	Tumor-like lesions
45	47	M	Trunk	115	0.5 h	0.3 mg/kg	2 h pre-op	yes	no	3	Malignancies	Fibrosarcoma
46	77	M	Pelvis	18	1 h	0.3 mg/kg	2 h pre-op	no	Yes	10	Malignancies	Fibrosarcoma
47	24	M	Femur	105	1 h	0.3 mg/kg	1.5 h pre-op	yes	no	10	Benign tumor	Chondroma
48	32	M	Trunk	71	1 h	0.3 mg/kg	2 h pre-op	yes	no	5	Malignancies	Ewing sarcoma
49	66	F	Trunk	25	1 h	0.3 mg/kg	1.5 h pre-op	yes	no	5	Malignancies	Metastases
50	61	M	Lower soft tissue	59	0.5 h	0.3 mg/kg	1.5 h pre-op	no	no	5	Benign tumor	Myxoma
51	11	F	Humerus	42	1 h	0.3 mg/kg	1.5 h pre-op	no	Yes	20	Malignancies	Osteosarcoma
52	47	M	Upper soft tissue	26	0.5 h	0.3 mg/kg	2 h pre-op	yes	no	5	Malignancies	Metastases
53	61	M	Lower soft tissue	105	1.5 h	0.3 mg/kg	0.5 h pre-op	yes	no	5	Malignancies	Liposarcoma
54	71	F	Femur	54	1 h	0.3 mg/kg	1 h pre-op	yes	no	10	Malignancies	Metastases
55	51	M	Lower soft tissue	65	1 h	0.3 mg/kg	0.5 h pre-op	yes	no	5	Benign tumor	Schwannoma
56	15	M	Lower soft tissue	57	1 h	0.3 mg/kg	1 h pre-op	yes	no	5	Malignancies	Liposarcoma
57	22	M	Lower soft tissue	61	1 h	0.3 mg/kg	1 h pre-op	yes	no	10	Malignancies	Liposarcoma
58	81	M	Trunk	66	1 h	0.3 mg/kg	0.5 h pre-op	yes	no	5	Malignancies	Metastases
59	48	M	Spinal column	51	1 h	0.3 mg/kg	1 h pre-op	yes	no	8	Benign tumor	Schwannoma

**Table 5 cancers-15-02402-t005:** Characteristics of 14 cases of first-time puncture specimens without real-time fluorescence development in the test group.

Number	Location	First Real-Time Fluorescence	Second Real-Time Fluorescence	First Biopsy Histopathology Diagnosis	Supplementary Biopsy Histopathology Diagnosis	After Surgery Resection Histopathology Diagnosis
1	Trunk	No	Yes	Non-tumor	Granulosa cell tumor	Granulosa cell tumor
2	Pelvis	No	Yes	Non-tumor	Metastatic adenocarcinoma	Metastatic adenocarcinoma
3	Pelvis	No	Yes	Non-tumor	Fibrosarcoma	Fibrosarcoma
4	Lower extremity	No	Yes	Non-tumor	Rhabdomyosarcoma	Rhabdomyosarcoma
5	Lower extremity	No	Yes	Non-tumor	Leiomyosarcoma	Leiomyosarcoma
6	Pelvis	No	Yes	Non-tumor	Fibrosarcoma	Fibrosarcoma
7	Humerus	No	Yes	Non-tumor	Osteosarcoma	Osteosarcoma
8	Trunk	No	No	Non-tumor	Non-tumor	Metastatic squamous cell carcinoma
9	Lower extremity	No	No	Non-tumor	Non-tumor	Liposarcoma
10	Spine	No	No	Non-tumor	Non-tumor	Metastatic adenocarcinoma
11	Femur	No	No	Tumor-like lesions	Tumor-like lesions	Tumor-like lesions
12	Pelvis	No	No	Tumor-like lesions	Tumor-like lesions	Tumor-like lesions
13	Tibia	No	No	Tumor-like lesions	Tumor-like lesions	Tumor-like lesions
14	Lower extremity	No	No	Tumor-like lesions	Tumor-like lesions	Tumor-like lesions

## Data Availability

Research data can be found at chictr.org.cn (accessed on 4 June 2022).

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
