# Peer review of "Application of Indocyanine Green Fluorescence Imaging in Assisting Biopsy of Musculoskeletal Tumors"

_cancers, 2023, doi:10.3390/cancers15082402_

Round 1
Reviewer 1 Report
The authors provided statistical analyses for the results. This point is significantly important.
Fig.3 should be shown before Fig.4.
In Fig.3 to Fig.5, please show the appropriate scale bars.
Fig.3D, Figs.4HM and Figs.5DI are not clear. Please show more clear images.
Author Response
Thank you for your review. I have modified the pictures in the article according to your requirements. Thank you again for your approval!
Reviewer 2 Report
Overall it is a very interesting manuscript with a novel approach to increase the diagnostic accuracy of soft tissue and bone tumors. The authors also highlighted the use of ICG and its impact in detecting tumor cells.
There are several issues in terms of the correct use of the English language and grammatical constructions, for example:
Simple Summary
The whole scope of the article is somewhat consuming due to some sentences being misconstructed and should be reviewed.
Abstract
The sentences are too long and confusing.
These are just a couple of examples, but throughout the manuscript there are similar issues that need to be addressed and edited
Regarding the scientific novelty, the authors presented a very interesting concept and developed a very “clean” protocol and should be commended by that.
I would like also to know how much the usage of the ICG and camera adds to the total cost of the procedure.
Table 4 should definitely be condensed as it is too big.
The final conclusions need to be reviewed as it seems overly ambitious to say that this procedure can or will facilite the surgery.
Author Response
Dear reviewer, I have modified the article according to your requirements. I have replied to your questions item by item in the attachment. Thank you again for your hard work.

Reviewer 3 Report
This represents a well written paper with a potentially exciting addendum to the literature. There are however some major issues to be addressed:
1.) ICG is proclaimed as the gold standard to guide the needle to well perfused areas of the heterogeneous tumor. It is even concluded that this increases the diagnostic yield:
a.) there was nowhere described/shown how this practically works. I urge the authors to show also infrared pictures and how based on these you then guide the needle. My point being:
-does the detection of the ICG is enough for deep seated lesions?
-how does -on your depicted imaging- help the 2D picture to guide the needle in the 3D of the tissues?
These two points are absolutely critical for this manuscript and need to be included/shown.
2.) there was no mentioning of the gold standard being the preop MRI-IVKM, where the areas of contrast uptake (indicating areas of perfusion) usually guide the needle to obtain most likely viable tissues. The authors have to show where these areas (MRI contrast uptake versus ICG accumulation) correlate.
If the authors will be unable to show this, then this paper cannot be accepted. Then, the addition of ICG injection is not improving the standard, which is still MRI with contrast. The main reason being: MRI imaging will always be necessary, and if the addition of ICG does not improve, the diagnostic yield of viable tissues, then it only adds costs and consumes time, which cannot be supported.
Further minor issues:
-please replace "puncture biopsy" with Core biopsy or fine needle biopsy. Use consistent throughout manuscript.
-please use an accepted MSKO biopsy classification (eg Mosque N et al, Cancers 2022
-Tables 1&2 can be omitted, do not add, statistics not really clear with these numbers.
-definition of "diagnostic performance" not really clear. The final path result is a culmination of various factors. One important parameter to include is the amount of necrosis measured (since this may complicate the biopsy if not retrieved from the correct anatomic spot)
-Table 5: 1st time fluorescence; does not add information, only complicates table.
-Table 4 please include amount of necrosis
-Figures do not show ICG in 3D and how the needle is placed accordingly.
Author Response
Dear reviewer, I have modified the article according to your requirements. And I have replied to your questions item by item in the attachment. Thank you again for your hard work.

Reviewer 4 Report
The manuscript described the utility of Indocyanine Green Fluorescence Imaging in assisting the biopsy of musculoskeletal tumors.
The topic is interesting and this may be useful in selected patients who need an accurate diagnosis.
1) Do the authors have any evidence for the positive reaction by Indocyanine Green in benign tumors? Is fluorescence always obtaIned in all benign tumors? What is the difference from normal tissue?
2) I think the tumor with homogeneous intensity on T2 MR image may be easy to obtain the sample. When the biopsy is planned in tumors that have heterogeneous intensity, this technique may be useful. How about the present cases?
3) Also, tumor size should be a critical factor for the difficulty of the accurate biopsy.
4) Also, the lytic or sclerotic change on CT should be a critical factor for the difficulty of the accurate biopsy. Please show it in the present cases.
5) We should consider the open biopsy depending on the cases. Please discuss it.
Author Response
Dear reviewer, I have modified the article according to your requirements. And I have replied to your requestions item by item in the attachment. Thank you for your hard work.

Round 2
Reviewer 3 Report
Thank you so much for the lengthy explanation. I did indeed misunderstood the set-up, and the fluorescence was assessed of the specimen ex vivo, as you have shown in the figures. However, even now reading the abstract, where nothing was changed, it is not clearly written that one understands this.
Having said all this, I am still asking what is the added value. The set-up would be much more logical if each of the retrieved specimen is directly correlated between fluorescence versus histology. I am not understanding why the authors argue that determination of necrosis is difficult in developing countries. The amount of necrosis is the absolute basic assessment in pathology, and everyone is capable to do that, it only needs HE staining. This doe snot make sense.
After all, and as it is now understood, this study shows that fluorescence can be shown in retrieved biopsied specimen, but it is not adjusted for how many biopsies were overall done for the patient, and being compared between control and test group.
Author Response

(The authors gave the same response as above.)

Reviewer 4 Report
I think the indication of this procedure should be described because all tumors may not be necessary. The authors replied in answer sheets about my queries, but they were not reflected in the manuscript. The authors mentioned the accuracy rate of percutaneous biopsy (69-88%). But, recently, superior outcomes were reported. (J Pediatr Hematol Oncol. 2014 Jul;36(5):333-6. Rofo. 2016 Dec;188(12):1156-1162.J Int Med Res. 2019 Jun;47(6):2598-2606.)
I think the present technique may be helpful in the selected patients. Therefore the authors should show the preferable cases.
Author Response
Dear reviewer, thank you for your review again. Your suggestion is of great value to our research, and I have revised the discussion section and added indication of this procedure in our discussion as you suggested. Using NIR imaging to assist biopsy is indeed most likely to be helpful for some selected patients, especially in tumors with heterogeneous in MRI. And I will continue our research to explore the indications you mentioned. At the same time, I have also made some changes to the conclusion. Thank you again for your timely review. I hope my revision will be approved by you.
Round 3
Reviewer 3 Report
I am impressed the work which was again put it into this revision. In my eyes, the perspectives are very well reflected now. Congratulations!
Reviewer 4 Report
Thank you for making the revision. I am satisfied with it.